# MAKE SVM GREAT AGAIN WITH SIAMESE KERNEL FOR FEW-SHOT LEARNING

## ABSTRACT

While deep neural networks have shown outstanding results in a wide range of applications, learning from a very limited number of examples is still a challenging task. Despite the difficulties of the few-shot learning, metric-learning techniques showed the potential of the neural networks for this task. While these methods perform well, they don't provide satisfactory results. In this work, the idea of metric-learning is extended with Support Vector Machines (SVM) working mechanism, which is well known for generalization capabilities on a small dataset. Furthermore, this paper presents an end-to-end learning framework for training adaptive kernel SVMs, which eliminates the problem of choosing a correct kernel and good features for SVMs. Next, the one-shot learning problem is redefined for audio signals. Then the model was tested on vision task (using Omniglot dataset) and speech task (using TIMIT dataset) as well. Actually, the algorithm using Omniglot dataset improved accuracy from 98.1% to 98.5% on the one-shot classification task and from 98.9% to 99.3% on the few-shot classification task.

## 1 INTRODUCTION

Deep learning has shown the ability to achieve outstanding results for real-world problems in various areas such as image, audio and natural language processing (LeCun et al., 2015). However these networks require large datasets, so the model fitting demands significant computational resources. On the other hand, there are techniques for learning on small datasets, such as data augmentation and special regularization methods, but these methods' accuracy is far from desirable on a very limited dataset. As well as slowness of the training process is caused by the many weight update iterations, which is required due to the parametric aspect of the model.

Humans are capable of learning the concept from only a few or even from one example. This learning characteristic differs much from the deep neural networks' learning curve. This discovery leads us to one-shot learning task (Fe-Fei et al., 2003), which consists of learning each class from only one example. Nevertheless, one single example is not always enough for humans to understand new concepts. In view of the previous fact, the generalization of one-shot learning task exists as well, it is called few-shot learning or $k$-shot learning, where the algorithm learns from exactly $k$ samples per class.

Deep learning approaches data-poor problems by doing transfer learning (Caruana, 1995): the parameters are optimized on a closely related data-rich problem and then the model is fine-tuned on the given data. In contrast, one-shot learning problem is extremely data-poor, but it requires similar approach as transfer learning: in order to learn good representation, the model is trained on similar data, where the classes are distinct from the one-shot dataset. In the next step, standard machine learning tools are used on the learned features to classify the one-shot samples. As a matter of fact, Vinyals et al. (2016) claimed that parameterless models perform the best, but they concentrated on only $k$-nearest neighbors algorithm. Considering this observation this work applies Support Vector Machine (Boser et al., 1992), which can be regarded as a parameterless model.

This paper presents the $k$-shot related former work in the following section. Then the proposed model, which is called Siamese kernel SVM, is introduced with a brief summary of the used well-known methods. In Section 4 the experimental setup is described for both a vision and an auditory task, where minor refinement of the problem is required.

## 2 RELATED WORK

The most obvious solution for the one-shot learning task is the $k$-nearest neighbors algorithm ($k$-NN). However, there is one problem with this algorithm, it requires complex feature engineering to work efficiently. When the number of available training data points is limited, Support Vector Machines are often used, as they generalize well using only a handful of examples, which makes them suitable for few-shot learning. The problem with SVMs is the same as with the $k$-nearest neighbors method: one must find set of descriptive features for a given task.

One of the neural network solutions for the one-shot learning problem is called Siamese network (Bromley et al., 1994), which relies on calculating pairwise similarities between data points. This architecture uses two instances of the same feedforward network to calculate representation before the similarity of the two observed samples are determined. Historically this architecture is created for verification problems, but it turned out that the model's learned representations can be used for classification tasks as well (Chopra et al., 2005). The first versions of Siamese networks used energy based, contrastive loss function (Chopra et al., 2005)
An improved version of the architecture is the Convolutional Siamese Net (Koch et al., 2015), which uses binary cross-entropy as loss function and convolutional network to learn features. Our work uses exactly the same convolutional architecture for vision task with a different loss function, which can learn better features for SVM classification. A different improvement of the Siamese architecture is the Triplet network Hoffer & Ailon (2015), which approaches the problem as a comparison of the data to a negative and a positive sample at the same time. This model uses three instances of the same feedforward networks: one for positive examples, one for negative examples and one for the investigated samples, which is put to the more similar class. One of the latest state-of-the-art models is Matching Network (Vinyals et al., 2016), which can be considered as an end-to-end $k$-nearest neighbors algorithm. This extension of the Siamese network contains $N + 1$ instances of the same network, where $N$ is the number of classes. The algorithm compares the sample to every classes' data points and chooses the class, which has the data points most similar to the investigated sample.

So far the distance metric learning approaches have been discussed, but there are different successful methods to solve the problem, such as Memory-Augmented Neural Network (Santoro et al., 2016), which is a Neural Turing Machine (Graves et al., 2014). It uses external memory to achieve good results in one-shot learning by memorizing the most descriptive samples. Another approach to the problem is meta-learning. Ravi & Larochelle (2016) use an LSTM (Hochreiter & Schmidhuber, 1997) based meta-learner that is trained to optimize the model's parameters for few-shot learning.

## 3 MODEL

### 3.1 SUPPORT VECTOR MACHINES

Linear Support Vector Machines (Linear SVM) are created for binary classification (Boser et al., 1992). Seeing that, the given data is labeled with +1 and -1: $\{(x_i, y_i)|x_i \in \mathbb{R}^D, y_i \in \{+1; -1\}\}$. Training of Linear SVMs are done by calculating the following constrained optimization, which is minimized with respect to $w \in \mathbb{R}^D$ as it is described in Equation 1.

$$\min_{w, \xi_i} \frac{1}{2} w^T w + C \sum_{i=1}^{N} \xi_i \tag{1}$$

$$\forall i : w^T x_i y_i \geq 1 - \xi_i$$
$$\forall i : \xi_i \geq 0$$

In Equation 1, $w^T w$ provides the maximal margin between different classes, which can be considered as a regularization technique. $\xi_i$-s are slack variables to create soft margin, they penalize data points inside the margin. Therefore, the C coefficient controls the amount of the regularization.

As SVMs are kernel machines, features of the data points are not required, only a positive-definite kernel is needed for training. Fortunately learned similarity metric is positive-definite. The SVM optimization problem's dual form makes it possible to optimize in kernel space, which may result in creating a nonlinear decision boundary. This means that during training only the kernel function is

required, which can be a precomputed Gram matrix. The dual form of the optimization problem has other useful properties: the training will find a sparse solution while the computational cost is lower if the number of training points is less than the number of features.

SVMs are binary classifiers, but they can be extended to multiclass classification with one-vs-rest method (Cortes & Vapnik, 1995). Although this paper investigates only the one-vs-rest approach, other methods are known for multiclass classification (Hsu & Lin, 2002) as well. The one-vs-rest approach can be interpreted as training N different SVMs (where N is the number of classes), each of which is used for deciding between given class and another. Equation 2 shows the prediction, where $w_k$ is k-th model's weight vector.

$$\arg\max_k(w_k^T x) \tag{2}$$

## 3.2 SIAMESE NETWORKS

Siamese network was first created for solving verification problem, where the data is given as $(x_1, x_2, y_{1,2})$, two samples and one label. Thus, the task is to predict, whether the $x_1$ example comes from the same class as the $x_2$ data point. The idea of Siamese network is to create a feedforward network in two instances with weight sharing, then construct a function to calculate the similarity or distance metric between the two instances (Bromley et al., 1994). The network's structure can be seen in Figure 1. The feedforward network does representation learning. Eventually, the similarity calculation can be a predefined function (Chopra et al., 2005) or it can be learned during the training (Koch et al., 2015) as well. The main requirements of the Siamese networks are:

- Siamese networks are *symmetric*. If two inputs are given in different order (($x_1, x_2$) or ($x_2, x_1$)), the result must be the same. This is provided via the similarity function.
- Siamese networks are *consistent* as well. Two very similar inputs are not projected to very different areas of the vector space. This is the consequence of the weight sharing.

Application of Siamese networks can be considered as a method for learning a similarity matrix (called Gram matrix) for all the possible pairs of samples.

Siamese networks can be used for classification too. Similarity can be transformed to distance, which is suitable for a $k$-NN classifier. This is the most popular solution for one-shot learning. Similarity matrix can be used by SVMs as we will see in Section3.3. Otherwise, representation of each instance can be used by any machine learning algorithm for classification.

## 3.3 SIAMESE KERNEL LEARNING FOR SVMS

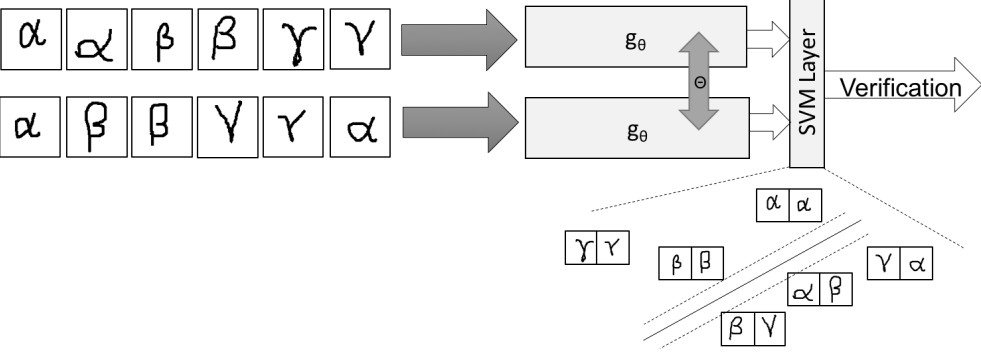

Figure 1: Verification Model: The network is fed with data pairs. $g_\Theta$ is a feature extractor function. Two instances of the $g_\Theta$ exist, the $\Theta$ parameter set is shared between instances. SVM layer separates same class pairs from different class pairs.

In the previous subsections, the two principal components of the model have been introduced. As Section 3.2 mentioned that Siamese networks were first trained on verification task. The verification

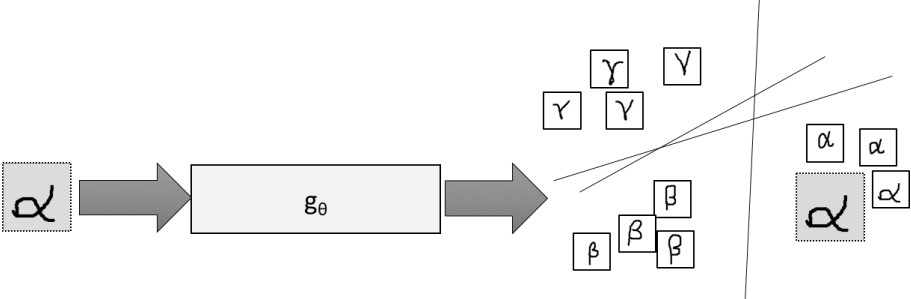

Figure 2: $g_\Theta$ is the feature extractor, which is the same $g_\Theta$ in the validation model. During $k$-shot learning, features of the data are calculated, and a linear multiclass SVM (figure shows one-vs-rest approach) is used with the same C parameter as the validation model's loss function.

architecture can be seen in Figure 1. The data preprocessing for the model is the same as Siamese network's process. Notably, the number of positive and negative samples are recommended to be equal. This is provided as all positive pairs are generated and the same amount of negative pairs are generated by choosing samples from different classes randomly. The negative sample generation is done on the fly, this can be considered as a mild augmentation.

In the verification architecture the SVM layer and its loss function have two parts:

- Equation 3 shows the feature difference calculation. Siamese network's symmetric attribution is provided via this function. Element-wise p-norm is perfect choice, this paper uses $L_1$ norm. In equation 3 n-th and m-th samples are compared, where $a^i$ refers to the vector's i-th element. This $\Phi^i_{n,m}$ is used by the SVM as input.

$$\forall i : \Phi^i_{n,m} = |g_\Theta(x_n)^i - g_\Theta(x_m)^i| \tag{3}$$

- This paper uses a popular version of linear SVMs, called L2-SVM, which minimizes squared hinge loss. Neural networks with different SVM loss functions are investigated in Tang (2013), L2-SVM loss variant is considered to be the best by the author of the paper. This loss function can be seen in Equation 4, where $y_{n,m} \in \{+1; -1\}$ is the label of the pair. The loss function's minimal solution is equivalent to the optimal solution of Equation 1.

$$\min_w \frac{1}{2} w^T w + C \sum_{n=1}^{N} max(1 - w^T \Phi_{n,m} y_{n,m}, 0)^2 \tag{4}$$

The used kernel is linear, so the data points' vectors in the SVM's feature space can be represented with finite dimension. Linear SVMs perform the best when data points in the SVM's feature space are separable by a hyperplane, which can be reached through high dimensional feature space. For this reason, a large number of neurons in $g_\Theta$'s last layer may increase performance when the number of classes is large. Another solution for increasing the feature space dimension is using a nonlinear kernel in the SVM Layer and in the loss function. For example, Radial Basis Function (RBF) kernel results in infinite dimension in the SVM's feature space. The main drawback of a nonlinear kernel is that the loss function must use the dual form of Support Vector Machine optimization (Boser et al., 1992), which can be computationally expensive in case of this architecture. Typically, the number of training samples is large for a deep model and the complexity of the gradient calculation for the last layer's weights in dual form is rather enormous. This computational complexity is $\mathcal{O}(m^2)$ indeed, where $m$ is the number of samples and all examples can be considered as a potential support vector. These gradients can be determined via dual coordinate descent method, which is analyzed in details in the Hsieh et al. (2008) article. In conclusion, the complexity of calculating the loss values for one epoch is $\mathcal{O}(n^4)$ due to the Siamese architecture's sample pair generation (considering batch size is independent of $n$), where $n$ is the number of samples. This makes the model hard to train on a large dataset. Yet, another problem of the dual form is that the number of parameters in the SVM Layer is equal to the number of samples, which makes the model tied to the dataset. Therefore, this paper investigates only linear SVM solutions due to this problem.

$K$-shot learning has two learning phases, the first is described as the verification learning phase, which is used for representation learning on an isolated dataset (see Figure 1). In the second phase, which is referred to as few-shot learning in this paper, the new classes are learned by a linear multi-class SVM (see Figure 2). The classifier model uses the representation of the data, which is provided by $g_\Theta$. The mentioned SVM has the same $C$ parameter as the squared hinge loss function, which is why the optimal representation for the Support Vector Machine is learned by $g_\Theta$. Therefore, this learning characteristic makes the neural network of an adaptive kernel for the SVM.

In a former paragraph, the possibility of a nonlinear kernel is investigated in the representation learning phase. This idea can be used in the second learning phase as well. However, the $g_\Theta$ function's output can not be used as an input data point of the nonlinear SVM because it is in its feature space. It is desired to use kernel space optimization instead. The verification network's output can be transformed to a valid kernel function. Hence, the Gram matrix can be generated by calculating the verification network's result for each pair. The SVMs can learn from a Gram matrix in kernel space, without features. This method can be used for linear kernel too, but the computational cost of this approach is larger because calculating the Gram matrix requires $\mathcal{O}(n^2)$ forward step in the neural network as it calculates all possible pairs while determining the pure features needs only $\mathcal{O}(n)$ forward step.

The described model is an end-to-end neural SVM, which has an adaptive kernel. In the next section, the model is used in several experiments on different datasets, then compared to the end-to-end K-NN model described in Vinyals et al. (2016).

## 4 EXPERIMENTS

### 4.1 OMNIGLOT

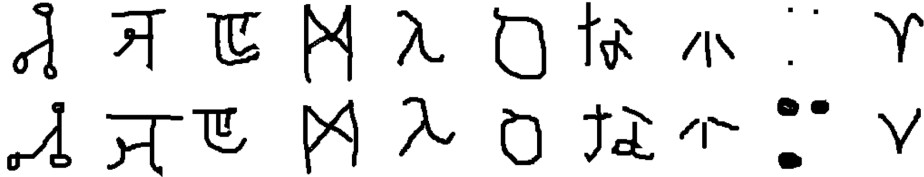

Figure 3: Omniglot dataset: all the characters are grayscale 105 x 105 images.

The Omniglot (Lake et al., 2011) dataset is a set of handwritten characters from different alphabets. Each character is written 20 times by different people. Furthermore, the total number of characters is 1623 and the characters come from 50 different alphabets. Figure 3 shows example images from the dataset. The dataset is collected via Amazon's Mechanical Turk.

The evaluation method is the same as described in Vinyals et al. (2016) paper, the models' accuracies on this dataset are shown in Table 1. For the experiment characters were mixed independently from alphabets: the first 1150 characters were used for training the kernel in order to learn representation, the next 50 characters were used for validation to select the best model. The remaining items were used for testing, where $n$ classes are chosen, and $k$ samples were used for training the SVM. It is called $n$-way $k$-shot learning. Each test was run 10 times using different classes to get robust results. During the training, no explicit data augmentation was used.

The used model's $g_\Theta$[1] is identical to Convolutional Siamese Network (Koch et al., 2015), which can be seen in Figure 4. The only difference is the regularization, the original model used L2 weight decay, while this model uses dropout layers(Srivastava et al., 2014) with 0.1 rates after every max pooling layer and before its last layer. The SVM's C parameter for regularization is 0.2. This model is trained for maximum 200 epochs with Adam optimizer (Kingma & Ba, 2014). Early stopping is used, which uses accuracy on "same or different class" task as stopping criteria. This slight modification in the training method results in big performance improvement as seen in Table 1.

---

[1]The code will be available soon.

The representation can be fine-tuned if the $k$-shot learning's training data is used for further fitting. This may result in massive overfitting and it can't be prevented with cross-validation in case of one-shot learning. During fine-tuning, the model is trained for 10 epochs. The data for fine-tuning is generated as described in Section 3.3. This can not be applied for one-shot learning, where the same class pairs don't exist. For this purpose, the pair is created from the original image and its augmented version.

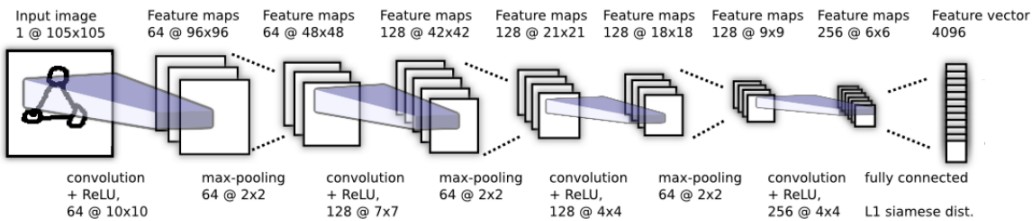

Figure 4: Feature extractor Convolutional neural network[2]

Table 1: Results on the Omniglot dataset [3]

| Model | Fine Tune | 5-way Acc | | 20-way Acc | |
| --- | --- | --- | --- | --- | --- |
| | | 1-shot | 5-shot | 1-shot | 5-shot |
| BASELINE CLASSIFIER | N | 80.0% | 95.0% | 69.5% | 89.1% |
| BASELINE CLASSIFIER | Y | 86.0% | 97.6% | 72.9% | 92.3% |
| CONVOLUTIONAL SIAMESE NET | N | 96.7% | 98.4% | 88.0% | 96.5% |
| CONVOLUTIONAL SIAMESE NET | Y | 97.3% | 98.4% | 88.1% | 97.0% |
| MATCHING NET | N | 98.1% | 98.9% | 93.8% | 98.5% |
| MATCHING NET | Y | 97.9% | 98.7% | 93.5% | 98.7% |
| NEURAL STATISTICIAN Edwards & Storkey (2016) | N | 98.1% | 99.5% | 93.2% | 98.1% |
| TCML Mishra et al. (2017) | N | **99.0%** | **99.8%** | **97.6%** | **99.4%** |
| PROTOTYPICAL NETWORKS Snell et al. (2017) | N | 98.8% | 99.7% | 96.0% | 98.9% |
| METANET Munkhdalai & Yu (2017) | N | **99.0%** | 98.7% | 97.1% | 97.0% |
| SIAMESE KERNEL SVM | N | 98.5% | 99.3% | 94.0% | 98.0% |
| SIAMESE KERNEL SVM | Y | 98.4% | 99.2% | 94.1% | 98.2% |

## 4.2 TIMIT

The task of one-shot learning is poorly defined on audio data because one-shot can be 1 second or even 5 seconds as well, therefore it is required to redefine the task. In this paper $k$-sec learning is defined so that the length of the training data is $k$ seconds regardless of the sample rate. Eventually,

---

[2]The figure originally published in Koch et al. (2015), this version contains minor modifications.

[3]The table contains results of Vinyals et al. (2016) regarding baseline classifier, Convolutional Siamese Net, and Matching Net models. The Siamese kernel SVM results are measured in the same experimental setup as other models

$w_{len}$ can be considered as a hyperparameter of the model, so optimal value of $w_{len}$ depends on the task as we will see. Furthermore, the length of each data point is exactly $w_{len}$ seconds, where $w_{len} \leq k$ is satisfied. In addition, these data points can partially overlap, but $k$ seconds length training data points mustn't overlap with evaluation points. Few seconds classification is an important task in real-world applications because it is exhausting to collect a large amount of data from speakers for robust classification. In this section, two scenarios are investigated: the first is a real-time application for speaker recognition, where $k$ is 1 second, this can be considered as the upper limit of the online recognition. The second case is where $k$ is 5 seconds, it is considered as an offline scenario.

TIMIT (Garofolo et al., 1993) is one of the most widely used English speech corpus. The dataset was originally designed for speech-to-text tasks. However, this dataset is perfect for speaker identification task too. The used dataset, which is projected from TIMIT contains audio files and their labels are the speakers. It contains 630 native speakers and the total number of sentences is 6300. Each speaker speaks for about 30 seconds. The official training set contains 462 people's voice. As a matter of fact, the training set is distinct from evaluation set regarding speakers, so neither of the training set speakers appears in the test set due to TIMIT is a speech-to-text task oriented dataset. This partitioning of the data makes the dataset unsuitable for a classical classification task, but it makes the TIMIT dataset perfect for $k$-sec learning task. There is no baseline known for $k$-sec learning problem on this dataset, so two different baseline models are introduced.

In this experiment, the official training set is used for training the models to learn representation and the chosen subsets of the evaluation set are used to train the model for the $k$-sec learning problem. The evaluation is the same as in the previous section, it is done on 10 different subsets.

For the neural models, the audio data is converted to a spectrogram, which can be handled as an image, see Figure 5.

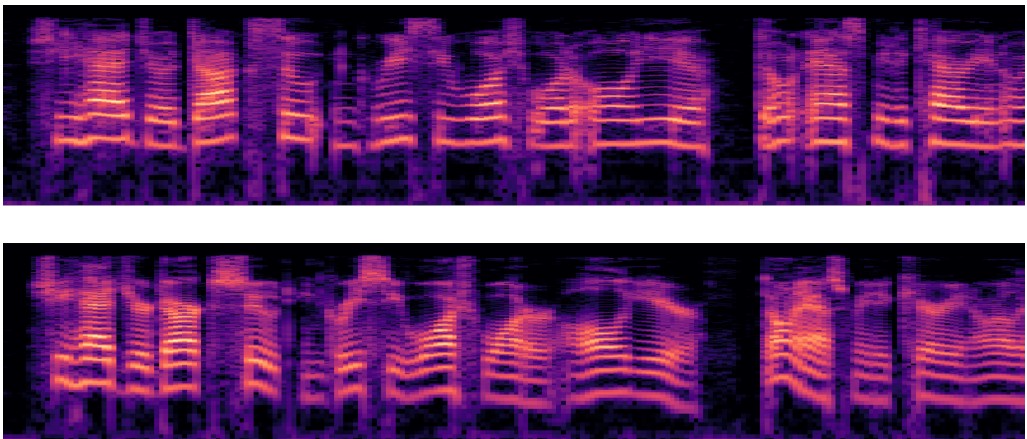

Figure 5: Spectrograms of two different people when the same text has been read. There are observable differences in tempo and tone as well.

Baseline models:

1. The first classifier uses handcrafted features with ensembles of SVMs. The used features are aggregated MFCC(Hunt et al., 1980) and LPCC(Waibel & Lee, 1990). This classifier has two versions for different length of training data. The first version is optimized for 1-sec learning, which uses 0.3 sec long audio with 0.1 sec offset sliding window. The second version is optimized for longer training data, which used 3 sec long slices and the sliding window steps by 0.1 sec.

2. The second model uses a neural network, which consists of convolutional layers (the architecture can be seen in Figure 6) and a fully connected layer on the top of the network. It is pretrained on the training set and the fully connected layer is changed to fit the problem then the model is fine-tuned for the chosen classes with transfer learning. The idea of using different window length for different tasks can be used here too. On the other hand, the re-

Table 2: Accuracies on the TIMIT dataset

| Model | Window length | 5-way Acc | | 20-way Acc | |
|---|---|---|---|---|---|
| | | 1-sec | 5-sec | 1-sec | 5-sec |
| BASELINE MODEL | 0.3 sec | 45.9% | 71.6 | 24.3% | 48.2% |
| (SVM-MFCC-LPCC) | 3.0 sec | - | 86.7% | - | 66.3% |
| BASELINE MODEL | 0.8 sec | 48.7% | 55.8% | 19.1% | 28.2% |
| (CNN-transfer learning) | 3.0 sec | - | 59.9% | - | 28.1% |
| SIAMESE KERNEL SVM | 0.8 sec | **86.1%** | 95.3% | **60.4%** | 84.2% |
| SIAMESE KERNEL SVM | 3.0 sec | - | **99.0%** | - | **95.9%** |

sults confirmed that classical neural network requires a lot of data, so can't take advantage of longer audio slices, if it causes fewer data point. The sliding window's offset here is 0.1 sec as well and the length of the window can be seen in Table 2.

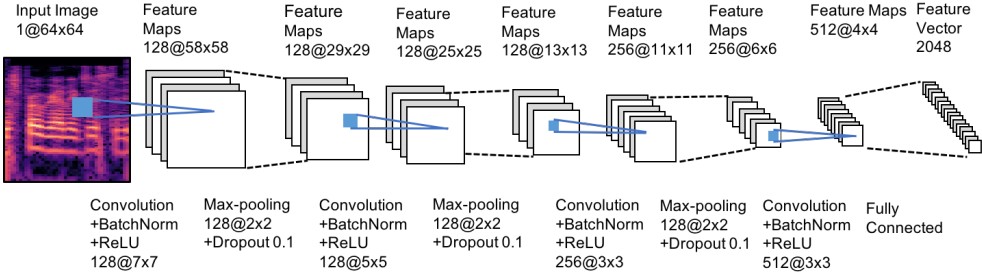

Figure 6: Feature extractor CNN for audio: used input's resolution is $64x64$ pixel.

The used neural SVM model's feature extractor can be seen in Figure 6. In the network, Batch Normalization layers(Ioffe & Szegedy, 2015) are used after every convolution layer in order to promote faster convergence and dropout layers (Srivastava et al., 2014) are also used with 0.1 rates after every max pooling layer to regularize the model. This model is trained for maximum 200 epochs with Adam optimizer (Kingma & Ba, 2014) and the best model has been selected with respect the same/different class accuracy for evaluation. The value of C is set to 15. As experiments with baseline models proved, optimizing sliding window length ($w_{len}$) to the task may significantly improve accuracy. During the spectrogram generation, $64x64$ pixel resolution images are created, which represents $w_{len}$ sec length audio, the exact value of $w_{len}$ in the experiments and the accuracies of the experiments can be seen in Table 2. Furthermore, a sliding window is used with 0.05 sec step size on the evaluation set, but 0.4 sec step size is used on training set due to computational complexity considerations.

The results (see Table 2) prove that the proposed task is complex enough for classical machine learning algorithms to not achieve satisfying accuracy and pure transfer learning not enough for suitable results. However, the proposed method's accuracy is far better than baselines. There is no major surprise, it is designed to perform well on a few data.

## 5  CONCLUSION

In this work, Siamese kernel SVM was introduced, which is capable of state-of-the-art performance on multiple domains on few-shot learning subject to accuracy. The key point of this model is combining Support Vector Machines' generalizing capabilities with Siamese networks one-shot learning

abilities, which can improve the combined model's results on the $k$-shot learning task. The main observation of this work is that learning representation for another model is much easier when the feature extractor is taught as an end-to-end version of the other model. In addition, parameterless models achieve the best results on the previously defined problem, which makes SVMs an adequate choice for the task. This paper also introduced the concept of $k$-sec learning, which can be used for audio and video recognition tasks, and it gave a baseline for this task on the TIMIT dataset. The author hopes defining $k$-sec learning task encourage others to measure one-shot learning models' accuracy on various domains.

This research focused on the linear kernel's case, but further improvements may be possible by investigating the use of nonlinear end-to-end SVM. The nonlinear setting's to be solved problems come overwhelmingly from the unreasonable amount of computational complexity of the optimizing the dual problem.

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
