# OpenReview forum: "Make SVM great again with Siamese kernel for  few-shot learning"
_ICLR.cc/2018/Conference — Reject_

### Official Review · AnonReviewer1 · 2017-11-23
**The paper is readable but it could be more fluent. It lacks a few references and important technical aspects are not discussed. It contains a few errors. Empirical contribution seems inflated on omniglot as the authors omit other papers reporting better results. Overall, the contribution is modest at best.**

**Rating:** 3
**Confidence:** 4

**Review:**

Make SVM great again with Siamese kernel for few-shot learning

** PAPER SUMMARY **

The author proposes to combine siamase networks with an SVM for pair classification. The proposed approach is evaluated on few shot learning tasks, on omniglot and timit.


** REVIEW SUMMARY **

The paper is readable but it could be more fluent. It lacks a few references and important technical aspects are not discussed. It contains a few errors. Empirical contribution seems inflated on omniglot as the authors omit other papers reporting better results. Overall, the contribution is modest at best.

** DETAILED REVIEW **

On mistakes, it is wrong to say that an SVM is a parameterless classifier. It is wrong to cite (Boser et al 92) for the soft-margin SVM. I think slack variables come from (Cortes et al 95). "consistent" has a specific definition in machine learning https://en.wikipedia.org/wiki/Consistent_estimator , you must use a different word in 3.2. You mention that a non linear SVM need a similarity measure, it actually need a positive definite kernel which has a specific definition, https://en.wikipedia.org/wiki/Positive-definite_kernel .

On incompleteness, it is not obvious how the classifier is used at test time. Could you explain how classes are predicted given a test problem? The setup of the experiments on TIMIT is extremely unclear. What are the class you are interested in? How many classes and examples does the testing problems have?

On clarity, I do not understand why you talk again about non-linear SVM in the last paragraph of 3.2. since you mention at the end of page 4 that you will only rely on linear SVMs for computational reasons. You need to mention explicitely somewhere that (w,\theta) are optimized jointly. The sentence "this paper investigates only the one versus rest approach" is confusing, as you have only two classes from the SVM perspective i.e. pairs (x1,x2) where both examples come from the same class and pairs (x1,x2) where they come from different class. So you use a binary SVM, not one versus rest. You need to find a better justification for using L2-SVM than "L2-SVM loss variant is considered to be the best by the author of the paper", did you try classical SVM and found them performing worse? Also could you motivate your choice for L1 norm as opposed to L2 in Eq 3?

On empirical evaluation, I already mentioned that it impossible to understand what the classification problem on TIMIT is. I suspect it might be speaker identification. So I will focus on the omniglot experiments.

Few-Shot Learning Through an Information Retrieval Lens, Eleni Triantafillou, Richard Zemel, Raquel Urtasun, NIPS 2017 [arxiv July'17]

and the reference therein give a few more recent baselines than your table. Some of the results are better than your approach. I am not sure why you do not evaluate on mini-imagenet as well as most work on few shot learning generally do. This dataset offers a clearer experimental setup than your TIMIT setting and has abundant published baseline results. Also, most work typically use omniglot as a proof of concept and consider mini-imagenet as a more challenging set.

---

### Official Review · AnonReviewer3 · 2017-11-24
**idea is reasonable but analysis is not very convincing**

**Rating:** 5
**Confidence:** 4

**Review:**

After reading the rebuttal:

This paper does have encouraging results. But as mentioned earlier, it still lacks systematic comparisons with existing (and strongest) baselines, and perhaps a better understanding the differences between approaches and the pros and cons. The writing also needs to be improved. So I think the paper is not ready for publication and my opinion remains.
===========================================================

This paper presents an algorithm for few shot learning. The idea is to first learn representation of data using the siamese networks architecture, which predicts if a pair of two samples are similar (e.g., from the same class) or not using a SVM hinge loss, and then finetune the classifier using few labeled examples (with possibly a different set of labels). I think the idea of representation learning using a somewhat artificial task makes sense in this setting.

I have several concerns for this submission.
1. I am not very familiar with the literature of few shot learning. I think a very related approach that learns the representation using pretty much the same information is the contrastive loss:
-- Hermann and Blunsom. Multilingual Distributed Representations without Word Alignment. ICLR 2014.
The intuition is similar: similar pairs shall have higher similarity in the learned representation, than dissimilar pairs, by a large margin. This approach is useful even when there is only weak supervision to provide the "similarity/dissimilarity" information. I wonder how does this approach compare with the proposed method.

2. The experiments are conducted on a small dataset OMNIGLOT and TIMIT. I do not understand why the compared methods are not consistently used in both experiments. Also, the experiment of speaker classification on TIMIT (where the inputs are audio segments with different durations and sampling frequency) is a quite nonstandard task; I do not have a sense of how challenging it is. It is not clear why CNN transfer learning (the authors did not give details about how it works) performs even worse than the non-deep baseline, yet the proposed method achieves very high accuracy. It would be nice to understand/visualize what information have been extracted in the representation learning phase.

3. Relatively minor: The writing of this paper is readable, but could be improved. It sometimes uses vague/nonstandard terminology ("parameterless") and statement. The term "siamese kernel" is not very informative: yes, you are learning new representations of data using DNNs, but this feature mapping does not have the properties of RKHS; also you are not solving the SVM dual problem as one typically does for kernel SVMs. In my opinion the introduction of SVM can be shortened, and more focuses can be put on related deep learning methods and few shot learning.

---

> ### Author Response · Authors · 2017-12-12
> **TIMIT results can be more clear**
>
> Thanks for the review.
> 1. We made experiments with contrastive loss and cross-entropy as well. We get the best results if the network is trained with a loss function, which is the same as the used classifier's loss (if it exists).
> In the OMNIGLOT setup, the same network was used to emphasize the importance of the loss function.
>
> 2. To our best knowledge the CNN transfer-learning doesn't enjoy the advantage of the pairwise data generation, so less data causes lack of generalization.
>
> 3. Agree with your point. This work focuses on linear SVMs, but we tried to give an outlook of the problem in Section 3.3, where we claimed kernel can be used and optimize the dual problem. This approach requires further research.

---

### Official Review · AnonReviewer2 · 2017-11-30
**No new idea and experiments are not convincing**

**Rating:** 4
**Confidence:** 5

**Review:**

Summary:
The paper proposes to pre-train a deep neural network to learn a similarity function and use the features obtained by this pre-trained network as input to an SVM model. The SVM is trained for the final classification task at hand using the last layer features of the deep network. The motivation behind all this is to learn the input features to the SVM as opposed to hand-crafting them, and use the generalization ability of the SVM to do well on tasks which have only a handful of training examples. The authors apply their technique to two datasets, namely, the Omniglot dataset and the TIMIT dataset and show that their model does a reasonable job in these two tasks.

While the paper is reasonably clearly written and easy to read I have a number of objections to it.

First, I did not see any novel idea presented in this paper. Lots of people have tried pre-training a neural network on auxiliary task(s) and using the features from it as input to the final SVM classifier. People have also specifically tried to train a siamese network and use its features as input to the SVM. These works go way back to the years 2005 - 2007, when deep learning was not called deep learning. Unless I have missed something completely, I did not see any novel idea proposed in this paper.

Second, the experiments are quite underwhelming and does not fully support the superiority claims of the proposed approach. For example, the authors compare their model against rather weak baselines. While the approach (as has been shown in the past) is very reasonable, I would have liked the experiments to be more thorough, with comparison to the state of the art models for the two datasets.

---

> ### Author Response · Authors · 2017-12-12
> **Idea: learn kernel of the problem instead of features.**
>
> Thanks for the review. In this paper, I wanted to point out that neural networks not only useful for feature learning, but kernel learning as well with proper techniques. Only in case of linear SVM, there is a shortcut: use the features.
> Can you provide a pointer to the mentioned prior work from 2005-2007?

---

### Decision · Program_Chairs · 2018-01-29
**ICLR 2018 Conference Acceptance Decision**

**Decision:**

Reject

**Comment:**

This paper proposes to pre-train a feature embedding, using Siamese networks, for use with few-shot learning for SVMs.  The idea is not very novel since there is a fairly large body of work in the general setting of pre-trained features + simple predictor.  In addition, the experimental results could be stronger -- there are stronger results in the literature (not cited), and better data sets for testing few-shot learning.